# Identification and Characterization of a Novel B Cell Epitope of ASFV Virulence Protein B125R Monoclonal Antibody

**DOI:** 10.3390/v16081257

**Published:** 2024-08-05

**Authors:** Yanyan Zhao, Haojie Ren, Zhizhao Lin, Saiyan Shi, Biao Zhang, Yuhang Zhang, Shichong Han, Wen-Rui He, Bo Wan, Man Hu, Gai-Ping Zhang

**Affiliations:** 1International Joint Research Center of National Animal Immunology, College of Veterinary Medicine, Henan Agricultural University, Zhengzhou 450046, China; 13592074304@163.com (Y.Z.); renhaojie1995@126.com (H.R.); 18855064740@163.com (Z.L.); 18749216585@163.com (S.S.); zhangbiao990112@163.com (B.Z.); zyh7125@163.com (Y.Z.); hanshichong081@126.com (S.H.); wrhe0111@163.com (W.-R.H.); wanboyi2000@163.com (B.W.); 2Longhu Laboratory, Zhengzhou 450046, China

**Keywords:** African swine fever virus, pB125R, monoclonal antibody, epitope identification

## Abstract

The African swine fever virus (ASFV) is an ancient, structurally complex, double-stranded DNA virus that causes African swine fever. Since its discovery in Kenya and Africa in 1921, no effective vaccine or antiviral strategy has been developed. Therefore, the selection of more suitable vaccines or antiviral targets is the top priority to solve the African swine fever virus problem. *B125R*, one of the virulence genes of ASFV, encodes a non-structural protein (pB125R), which is important in ASFV infection. However, the epitope of pB125R is not well characterized at present. We observed that pB125R is specifically recognized by inactivated ASFV-positive sera, suggesting that it has the potential to act as a protective antigen against ASFV infection. Elucidation of the antigenic epitope within pB125R could facilitate the development of an epitope-based vaccine targeting ASFV. In this study, two strains of monoclonal antibodies (mAbs) against pB125R were produced by using the B cell hybridoma technique, named 9G11 and 15A9. The antigenic epitope recognized by mAb 9G11 was precisely located by using a series of truncated ASFV pB125R. The ^52^DPLASQRDIYY^62^ (epitope on ASFV pB125R) was the smallest epitope recognized by mAb 9G11 and this epitope was highly conserved among different strains. The key amino acid sites were identified as D52, Q57, R58, and Y62 by the single-point mutation of 11 amino acids of the epitope by alanine scanning. In addition, the immunological effects of the epitope (pB125R-DY) against 9G11 were evaluated in mice, and the results showed that both full-length pB125R and the epitope pB125R-DY could induce effective humoral and cellular immune responses in mice. The mAbs obtained in this study reacted with the eukaryotic-expressed antigen proteins and the PAM cell samples infected with ASFV, indicating that the mAb can be used as a good tool for the detection of ASFV antigen infection. The B cell epitopes identified in this study provide a fundamental basis for the research and development of epitope-based vaccines against ASFV.

## 1. Introduction

African swine fever (ASF) is a hemorrhagic disease caused by the African swine fever virus (ASFV). The infection causes clinical symptoms such as high fever, rapid heartbeat, skin cyanosis, and hemorrhagic lesions in domestic pigs and wild boars. This disease is highly infectious and has a fatality rate of approximately 100% [1,2]. Since its first identification in Kenya in 1921, ASF has spread to most sub-Saharan African countries. In August 2018, ASF was first detected in China at a pig farm in Shenyang, Liaoning Province [1]. Subsequently, the rapid spread of ASF across China has devastated the pig industry, resulting in substantial economic losses [3]. ASF has also spread to other Asian countries, including Vietnam, Cambodia, Mongolia, Myanmar, Laos, the Philippines, North Korea, and South Korea [4,5,6].

ASFV is an enveloped double-stranded DNA virus. The complete ASFV virion consists of five layers: the nucleoid, nucleocapsid, inner membrane, icosahedral capsid, and outer membrane [7]. The genome length ranges from 170 to 193 kb depending on the isolate, encoding 150 to 167 proteins involved in viral replication, transcription, and pathogenesis [8,9,10,11]. Owing to the complexity of the virus’ structure and the limited understanding of protective immune mechanisms, efforts to develop an effective vaccine have been unsuccessful [11,12]. However, subunit vaccines have great potential. These vaccines are typically prepared using purified recombinant proteins or recombinant proteins with specific epitopes. Recombinant proteins (p30, p54, and p72) are prime targets for developing subunit vaccines and are commonly used in immunoassays to activate humoral and cellular immunity in immunized animals. Previous studies have used the enzyme-linked immunospot method to identify the epitope of ASFV T cells, providing a basis for vaccine development [13,14,15].

Nonstructural proteins also play important roles in ASFV infection and replication. Previous studies have reported that pB602L, a non-structural protein encoded by ASFV, is a companion for the main capsid protein, p72 [16,17]. Furthermore, the pB602L protein has good antigenicity, and antibodies against the pB602L protein can be detected as early as ten days after inoculation in pigs [18]. ASFV pB125R is an important non-structural protein as well, but its function is still unknown; thus, determining the antigenicity of pB125R is necessary to clarify its role.

Therefore, this study aimed to identify the monoclonal antibodies that target the ASFV pB125R protein. The antigenic, structural, and functional properties and immune activity of pB125R were assessed by identifying their epitopes. The overall goal in this study was to elucidate the function of the pB125R protein during ASFV infection, and, further, to provide a basis for the development of ASFV epitope vaccines.

## 2. Materials and Methods

### 2.1. Cells and Reagents

HEK293T cells were provided by Professor Hong-Bing Shu, and PK-15 cells were preserved in the laboratory. They were all cultured in Dulbecco’s modified Eagle’s medium (Solarbio, Beijing, China) supplemented with 10% fetal bovine serum (Gibco/Thermo Fisher Scientific, Waltham, MA, USA) at 37 °C in 5% CO_2_. Mouse myeloma cells (SP2/0) were obtained from the American Type Culture Collection (Manassas, VA, USA) and cultured in hybridoma cell serum-free medium (Basal Media, Shanghai, China) at 37 °C with 5% CO_2_. Competent DH5α and DE3 Escherichia coli cells were purchased from Tsingke Biotechnology (Beijing, China).

The PCMV-ASFV-pB125R plasmid was maintained in our laboratory. Antibodies against Flag (66008-4-Ig) or His (66005-1-Ig), and a goat anti-mouse IgG secondary antibody coupled with horseradish peroxidase (HRP; SA00001-1) were purchased from Proteintech (Wuhan, China). ASFV-positive serum was ordered from the China Veterinary Drug Administration (Beijing, China). Fluorescein DyLight 488 Goat Anti-Mouse immunoglobin IgG (A23210) was purchased from AbbKine (Beijing, China). PE anti-mouse CD3, FITC anti-mouse CD4, and APC-A700 anti-mouse CD8 were from proteintech (Wuhan, China). p72 mAb was provided by the Key Laboratory of Animal Immunology, Henan Academy of Agricultural Sciences (Zhengzhou, China). DAPI (C0065) was purchased from Solar Bio (Beijing, China). The protein samples of ASFV-infected PAM cells were provided by the Harbin Veterinary Research Institute.

The Ni-Agarose His Purification Kit for protein purification was purchased from CoWin Biosciences (Beijing, China). The complete and incomplete adjuvants required for immunization were ordered from Sigma-Aldrich (St. Louis, MO, USA). The following media were used for screening: polyethylene glycol 1450 (PEG1450; Sigma-Aldrich), hypoxanthine-aminopterin thymidine (HAT; Sigma-Aldrich), and hypoxanthine-thymidine medium (HT; Sigma-Aldrich). A mouse monoclonal antibody-isotyping ELISA kit (Proteintech) and tetramethylbenzidine (TMB, Solarbio) were used to identify the mAbs.

### 2.2. Preparation of Immune Serums

The animals involved in this study were immunized by subcutaneous immunization, and the purified biologically active recombinant prokaryotic protein was injected into mice subcutaneously after sufficient emulsification with the immune adjuvant. Mouse venous blood was collected from the tail tip at the time needed for the experiment, and then left at 37 °C for 30 min. After the blood coagulated and the blood clots contracted and precipitated the serum, it was centrifuged at 3000 rpm for 15 min, and the supernatant was taken, i.e., the desired serum was obtained, and then stored in the refrigerator at −80 °C.

### 2.3. pB125R Prokaryotic Expression Vector Construction

The reference sequence of ASFV B125R (GenBank: MK128995.1) was obtained from the National Center for Biotechnology Information database. Specific primers were designed (Table 1), and the amplified target gene was inserted into the pET-32a vector through the *Eco*R I and *Xho* I restriction enzyme sites. Then, the constructed plasmids were transformed into DH5α and verified by DNA sequencing (Sun Ya, Shanghai, China) to obtain the correct recombinant plasmid, named pET-32a-ASFV-pB125R.

### 2.4. pB125R Protein Expression and Purification

To obtain the prokaryotic pB125R protein, the recombinant plasmid pET-32a-ASFV-pB125R was transformed into E. coli strain DE3. The recombinant His-tagged protein was induced by Isopropyl *β*-D-1-thiogalactopyranoside (IPTG) (0.8 mmol/L) for 8 h. Supernatant and precipitate were collected after high-pressure homogenization. The recombinant His-tagged protein was identified by sodium dodecyl sulfate-polyacrylamide gel electrophoresis (SDS-PAGE). The recombinant protein pB125R was purified using a Ni-Agarose His Purification kit and dialyzed by a phosphate buffer solution.

### 2.5. Production of mAbs against the ASFV pB125R Protein

An amount of 10 μg of purified pB125R, after being emulsified with Freund’s complete/incomplete adjuvant (Sigma-Aldrich, USA), was injected into each BALB/c female mouse, aged 6–8 weeks. Booster immunization was performed on mice at 14, 28, and 42 days after the first immunization. The antibody titers in the serum of mice were detected at 42 days by ELISA. The mice with the highest antibody titers were selected for hyper-immunization with 30 μg of pB125R protein. Three days later, the mice were killed, and the spleen cells of the mice were fused with Sp2/0 cells using 50% (*w*/*v*) PEG1420. The fused cells were screened in 96-well plates with HAT and cultured in HT medium for 7 days. Positive cell lines were screened by indirect ELISA. At last, a mouse monoclonal antibody isotype ELISA kit was used to identify the antibody subtypes.

### 2.6. Indirect ELISA

Serum antibody titers of the immunized mice were detected by indirect ELISA to screen positive hybridoma cells. The 96-well ELISA plates were coated with recombinant pB125R protein (200 ng/well) at 4 °C for 20 h. The plates were washed twice with phosphate-buffered saline with Tween (PBST) and blocked with the blocking buffer, the PBST solution containing 1.5% bovine serum albumin (BSA), at 37 °C for 2 h. After washing four times with PBST, the supernatant of the hybridoma cells was added to the wells and incubated at 37 °C for 1 h. After washing, a 5000-fold dilution of HRP-coupled goat anti-mouse IgG was added and incubated at 37 °C for 1 h. Finally, after washing five times with PBST, TMB substrate was added to the plate, the color was allowed to develop for 5 min, and then 2 M H_2_SO_4_ was added to terminate the reaction. The absorbance values at OD_450nm_ were measured using an ELISA orifice plate reader (Gallop Chemical Equipment and Technology Co., Ltd., Beijing, China).

### 2.7. Western Blotting

Western blotting was used to identify the expression of the target protein and the reactivity between different antigens and mAbs. The prokaryotic-expressed pB125R protein, truncated pB125R, and the eukaryotic-expressed pB125R epitope mutant proteins were isolated by 12.5% SDS-PAGE and transferred to a polyvinylidene fluoride (PVDF) membrane. After blocking with the PBST solution containing 5% skim milk powder at 37 °C for 30 min, the PVDF membrane was incubated with the indicator antibody at 37 °C for 1 h. Finally, film exposure was used to image the protein-containing membranes. The electrophoretic gel of the prokaryotic proteins was stained with Coomassie bright blue solution for 2 h. After decolorization, the scanned electrophoretic bands were retained.

### 2.8. Immunofluorescence Assay (IFA)

The PCMV-ASFV-pB125R plasmid was transfected into the PK-15 cells. Approximately 20 h after transfection, the cells were fixed with 4% paraformaldehyde for 30 min and then blocked with 5% BSA solution for 30 min. Then, the cells were incubated with mouse anti-Flag mAb or the positive hybridoma cell supernatant at 37 °C for 1 h. After washing five times with PBST, the cells were incubated with the fluorescein DyLight 488 Goat Anti-Mouse IgG at 37 °C for another 1 h. After staining with DAPI, the cells were imaged using a fluorescence microscope (Olympus, Tokyo, Japan).

### 2.9. Antigenic Epitope Identification

Table 2 lists the primers used in this study. The primers were designed to obtain the truncated target genes by polymerase chain reaction (PCR). The gene fragments were attached to the pET-32a vector, respectively, to obtain the truncated prokaryotic expression plasmid of ASFV pB125R. After SDS-PAGE and protein immunoassay, a series of single amino acid mutants of the pB125R protein were constructed on the PCMV eukaryotic expression vector, respectively, and the key amino acids of pB125R bound to mAb 9G11 were assessed by Western blotting. The pB125R truncated protein amino acid sequence is shown in Table 3.

### 2.10. Bioinformatic Analysis of the pB125R Protein

To analyze the genetic specificity of the ASFV pB125R protein epitopes, MEGA7 was used to compare the pB125R protein sequences from different ASFV strains. Table 4 lists the reference sequences of the different ASFV strains. I-TASSER (https://seq2fun.dcmb.med.umich.edu//I-TASSER/, accessed on 4 July 2023) was used to predict the three-dimensional (3D) structure of the pB125R protein to elucidate its spatial position.

### 2.11. Immunogenicity of the Epitope of pB125R

The selected epitope of pB125R, named pB125R-DY (^52^DPLASQRDIYY^62^), was purified and used to immunize Kunming mice. Fifteen Kunming mice aged 6–8 weeks were divided equally into three groups, called the phosphate-buffered saline (PBS) group, pB125R protein group, and PB125R-DY protein group. The levels of IgG, IgG1, and IgG2a antibodies in the serum of mice in each group were detected by indirect ELISA at 7, 14, 28, and 35 days. Splenic T-lymphocyte differentiation was detected by flow cytometry at 49 days after the first immunization.

### 2.12. Flow Cytometry

Splenic lymphocytes were separated and counted. Then, 10^6^/mL of splenic lymphocytes (10^6^ cells/mL) were incubated with PE-labeled anti-CD3 mAb and FITC-labeled anti-CD4 mAb or PE-labeled anti-CD3 mAb and APC-A700 labeled anti-CD8 mAb at 4 °C for 30 min, and fluorescence changes were detected by flow cytometry. Then, the distribution of CD4+ and CD8+ T cells was evaluated.

## 3. Results

### 3.1. The Expression and Purification of ASFV pB125R Recombinant Protein

After amplification by PCR, the ASFV pB125R gene was inserted into the pET-32a vector, then transformed the recombinant plasmid into the *E. coli* DH5α competent cells, and the plasmid with successful insertion of the gene fragment was referred to as pET-32a-ASFV-pB125R. Next, the pET-32a-ASFV-pB125R was transformed into the E. coli DE3 competent cells, and the pB125R protein was successfully expressed by IPTG induction. The recombinant pB125R protein was purified by Ni-Agarose His Purification Kit, and the results of SDS-PAGE showed that the purified pB125R protein was successfully obtained (Figure 1A). The results of Western blotting demonstrated that the purified protein could be recognized by both His-labeled antibodies and ASFV-positive serum (Figure 1B,C). These results indicated that the purified recombinant protein pB125R had good immunogenicity and could be used as an immunogen to immunize mice to prepare mAbs against pB125R.

### 3.2. Antibody Titers of the Serum from Mice Immunized with pB125R Protein

The mice were immunized four times with 10 μg recombinant pB125R protein, and the interval between each immunization was two weeks. The serum of each mouse was collected at 14, 28, and 42 days after the first immunization. Antibody titers of the serum from mice were measured by indirect ELISA, and the results showed that all antibody titers were above 1/25,600 (Figure 2A) at 42 days after the first immunization. Then, the eukaryotic vector PCMV-ASFV-pB125R was transfected into HEK293T cells, and the recombinant Flag-pB125R protein was successfully expressed (Figure 2B). As expected, the eukaryotic recombinant protein could be recognized by the antiserum of the immunized mice (Figure 2C), indicating that the immunized mice could be used for subsequent mAb preparation.

### 3.3. Preparation and Identification of mAbs against the pB125R Recombinant Protein

Mice with the highest serum antibody titer were subjected to a booster immunization with 30 μg recombinant pB125R protein. Three days later, the splenocytes were collected from the mice and fused with myeloma cells SP2/0 by 50% (*w*/*v*) PEG1420. Hybridoma cells generated after cell fusion were cultured in 96-well plates and cloned for 10 days. Indirect ELISA was used to screen for positive hybridoma cells, and after three rounds of cloning, two strains of anti-pB125R mAbs, 9G11 and 15A9, were obtained (Figure 3A). A mouse monoclonal antibody isotype ELISA kit was used to identify the antibody subtypes. The results showed that the heavy chain type of mAb 9G11 was identified as IgG2b, and the heavy chain type of mAb 15A9 was identified as IgG1. The light chain subtypes of mAb 9G11 and 15A9 were both kappa (Figure 3B).

To further confirm the immunogenicity of these two monoclonal antibodies, eukaryotic-expressed recombinant Flag-pB125R protein was prepared, and the reactions between the mAb and Flag-pB125R were verified by IFA and Western blotting. The results showed that both mAbs 9G11 and 15A9 could react with the eukaryotic-expressed pB125R protein. Moreover, the sensitivity of mAb 9G11 was relatively high (Figure 3C,D). Similarly, after infection with ASFV, proteins expressed in PAM cells could be recognized by both mAbs, and mAb 9G11 exhibited higher binding activity (Figure 3E). In summary, mAb 9G11 with better binding activity should be selected for further study, which will help to provide support for the research of the treatment and diagnosis of ASF.

### 3.4. Mapping mAb 9G11epitopes

Since mAb 9G11 showed better responsiveness to protein pB125R, we sought to identify the specific antigenic epitopes recognized by mAb 9G11. DNASTAR version 7.1 software (DNASTAR Inc., Madison, WI, USA) was used to predict the secondary structure of pB125R (Figure 4A). Four overlapping PCR fragments covering the entire pB125R gene were prepared by analyzing the amino acid sequences of the protein and subcloning them into the pET-32a expression vector. The recombinant plasmids (named pB125R-1(1-40AA), pB125R-2(30-70AA), pB125R-3(60-100AA), and pB125R-4(90-125AA)) were verified by sequencing, and the expression was then induced in DE3 *E. coli* cells. SDS-PAGE and Western blotting showed that pB125R-2(30-70AA) could react with mAb 9G11 (Figure 4B). To further determine the epitope of the protein recognized by mAb 9G11, pB125R-2(30-70AA) was cut into five segments, and their expression was induced. The results indicated that pB125R-2-4(51-65AA) and pB125R-2-5(52-64AA) could react with mAb 9G11, with pB125R-2-5(52-64AA) showing the most significant reaction (Figure 4C). This may be related to the recognition characteristics of the monoclonal antibody, and the 52-64AA sequence with better binding activity of mAb 9G11 was selected for further study.

For more accurate localization, pB125R-2-5(52-64AA) was sequentially truncated and then expressed in DE3 E. coli cells. Western blotting results revealed that the smallest epitope recognized by mAb 9G11 was pB125R-2-5(52-62AA) (Figure 4D), and D52 and Y62 of pB125R protein were important for the recognition of mAb 9G11. Next, to identify the other key amino acids of the pB125R protein recognized by mAb 9G11, a series of single amino acid mutants of pB125R-2-5(52-62AA) was constructed and expressed in a PCMV eukaryotic vector. Western blotting results showed that in addition to recognizing D52, Q57, R58, and Y62 were also key sites for the recognition of pB125R protein by mAb 9G11 (Figure 4E). After that, we tested whether the ASFV-positive serum could react with the epitope recognized by mAb 9G11. Western blotting results showed that the ASFV-positive serum showed good reactivity with the screened epitope protein (Figure 4F).

### 3.5. Bioinformatics Analysis of the Epitope in Different ASFV Strains

To determine the conservation of ASFV pB125R in different countries and regions, we compared the protein sequences of pB125R in different ASFV strains in GenBank. The epitopes were highly conserved in different ASFV pB125R strains, and the sequence similarity of epitopes 52-62AA (mAb 9G11 recognition) in different ASFV pB125R strains was 100% (Figure 5A). I-TASSER was used to predict the 3D structural model of ASFV pB125R, which was displayed by PyMol software (PyMol 2.0; available at: http://www.pymol.org/pymol, accessed on 4 July 2023). The results showed that epitope ^52^DPLASQRDIYY^62^ identified by mAb 9G11 was located on the lateral surface of the pB125R protein, and the key sites identified by mAb 9G11 were displayed symmetrically at both ends and the center of the epitope. Thus, epitope ^52^DPLASQRDIYY^62^ may be a linear epitope, and its key amino acids may play a role in binding to other ligands (Figure 5B).

### 3.6. Verification of the Epitope Immunoactivity

To verify the immunoactivity of the epitope ^52^DPLASQRDIYY^62^, the recombinant epitope protein pB125R-DY was expressed and purified (Figure 6A). Mice were immunized with purified pB125R-DY, and serum was collected at 7, 14, 28, and 35 days after the first immunization for antibody detection. As expected, both pB125R-DY and the full-length pB125R protein could induce elevated levels of serum IgG, IgG1, and IgG2a antibodies in mice compared to negative controls (Figure 6B). Thirty-five days after the first immunization, lymphocytes were isolated from the spleens of mice, and T cell subsets were determined by flow cytometry. The results showed that both pB125R-DY and the full-length pB125R protein had the same function, stimulating a significant increase in CD4+ and CD8+ T cells (Figure 6C).

## 4. Discussion

ASF is a potent infectious disease of swine with high lethality, posing a great threat to the economy and society since its outbreak. However, due to the complex structure of ASFV, most of the gene functions are unknown, which seriously limits the research on diagnosis and treatment methods and novel vaccines. Therefore, it is extremely necessary to explore the unknown functional proteins of ASFV.

The ASFV virulence gene is very important for the development of vaccines, and *I177L*, *CD2v*, and *H108R* are several important virulence genes of ASFV, and their deletion could weaken the pathogenicity after entering the animal, which could reduce the clinical symptoms of ASFV infection to a certain extent and protect the animal from infection with the parental strain [19,20,21]. Previous studies have shown that ASFV virulence gene *B125R* is a late-expressed gene, encoding a non-structural protein that plays an important role in the process of viral infection, and its deletion would weaken the virulence of the virus in domestic pigs [22]. At the same time, our study showed that the B125R gene encoding pB125R could be recognized by ASFV-positive serum and had good immunogenicity, which laid a foundation for the preparation of an ASFV subunit vaccine.

Compared with traditional vaccines, epitope-based vaccines have better security and are better able to cope with mutant strains of the virus. Epitope screening is the basis and premise of constructing epitope vaccines. ASFV p72, p54, and CD2v are considered to be key targets for vaccine development. Currently, B-cell linear epitopes have been identified on the basis of mAb, and these molecular epitopes will also become important targets for open-position vaccines [23,24]. A tandem epitope vaccine is the combination of dominant epitopes of an antigen in series to improve immune effects and reduce adverse immune reactions. In one study, a nanovaccine prepared by covalently linking dominant B cell and T cell epitopes of ASFV antigens (p72, CD2v, pB602L, and p30) induced a stronger T cell and B cell immune response in mice [25]. In our study, we identified a B-cell linear epitope 52–62 aa, highly conserved and located on the protein surface, which could effectively induce certain humoral and cellular immunity in a mouse model, providing a new target for the design and development of ASFV epitope vaccines. At the same time, we identified several key amino acids that bind the epitope to mAb by mutating a single amino acid of the epitope, providing a basis for the development of tandem epitope vaccines.

In this study, pB125R could be used as a candidate protective antigen, and the epitope of its antibody generation was identified, providing a basis for the preparation of tandem epitope vaccines. At the same time, the prepared pB125R mAbs also provided materials for the investigation of the functional mechanism of pB125R.

## 5. Conclusions

This study prepared two strains of anti-ASFV pB125R monoclonal antibodies and found a new B-cell epitope ^52^DPLASQRDIYY^62^ against the monoclonal antibody 9G11. In addition, this epitope was highly conserved in different ASFV strains. The epitope could also induce humoral and cellular immunity in a host. Overall, these two mAb strains and the validated targeted epitope protein of 9G11 provide a tool for studying the mechanisms of viral infection and developing epitope vaccines.

## Figures and Tables

**Figure 1 viruses-16-01257-f001:**
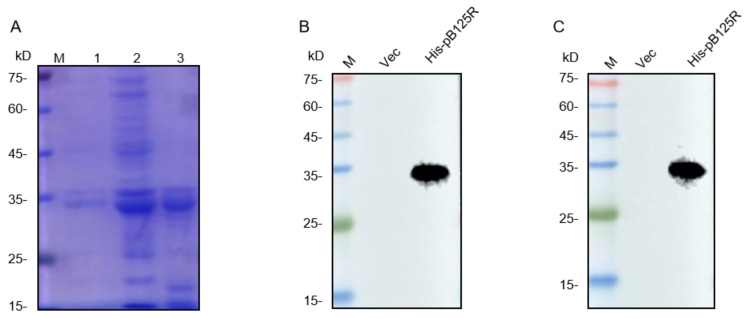
Preparation and purification of pB125R protein. (**A**) SDS-PAGE analysis of the expression and purification of recombinant pB125R protein. Lane 1: pB125R was not induced; lane 2: pB125R with IPTG; lane 3: purified pB125R. (**B**) Western blotting analysis of recombinant pB125R protein using anti-His antibodies. (**C**) Western blotting analysis of recombinant pB125R protein using ASFV-positive serum. M: Protein molecular weight marker.

**Figure 2 viruses-16-01257-f002:**
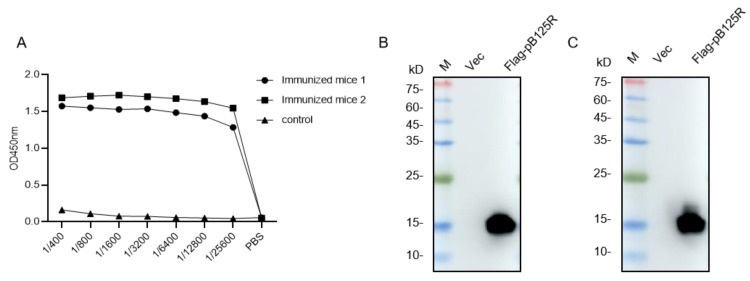
Identification of immunogenicity of pB125R protein. (**A**) The serum titer of mice was measured by indirect ELISA. Control: negative mice immunized with PBS. (**B**) Flag antibody was used for Western blotting analysis to detect the eukaryotic expression of recombinant pB125R protein. (**C**) The eukaryotic expression of recombinant pB125R protein was detected by Western blotting in the antiserum of immunized mice. Primary antibody: the serum of Mouse 2# with the highest titer. M: protein molecular weight marker.

**Figure 3 viruses-16-01257-f003:**
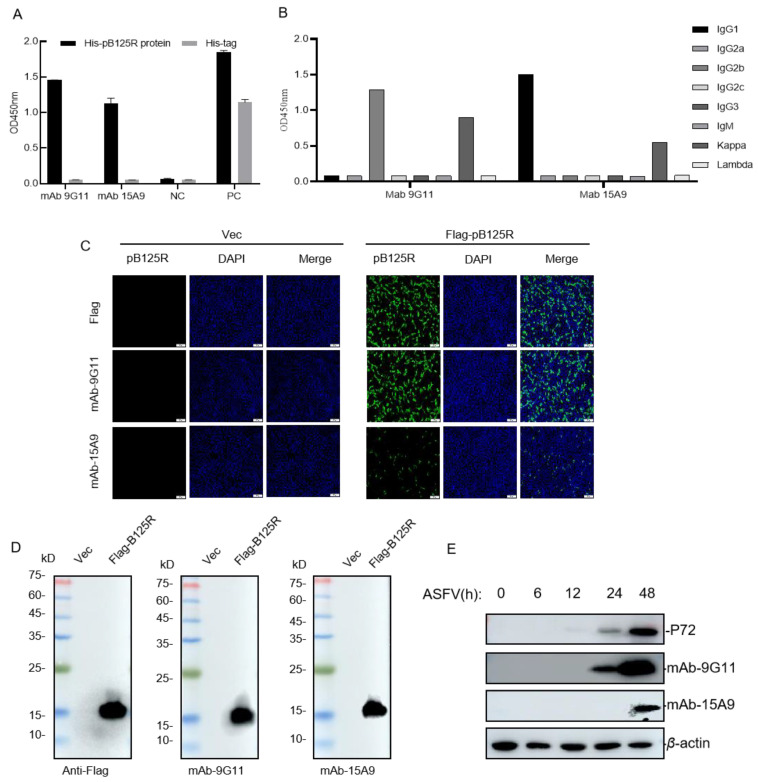
Identification of pB125R mAbs. (**A**) mAb titer was determined by indirect ELISA. (**B**) Identification of pB125R mAbs subtype. (**C**) The reactivity of mAbs was analyzed by IFA. (**D**) Western blotting assay to identify the reactivity of mAbs to eukaryotic-expressed pB125R. (**E**) The reactivity of mAbs to virus pB125R was analyzed by Western blotting.

**Figure 4 viruses-16-01257-f004:**
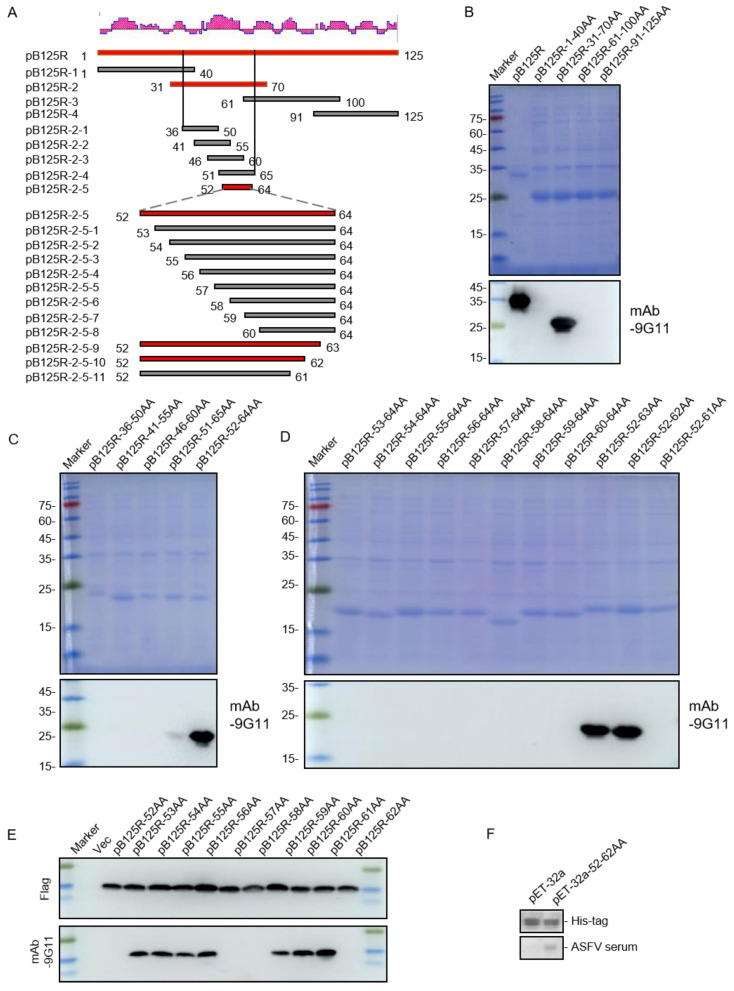
ASFV pB125R epitope identification. The pB125R gene is divided into overlapping segments. It was expressed with a pET-32a vector and detected with screened mAb 9G11. (**A**) Epitope location diagram. Fragments that are recognized by mAb 9G11 are highlighted in red. Gray: Amino acid sequence that does not react with mAb 9G11; Red: Amino acid sequence reacting with mAb 9G11. (**B**) Truncated ASFV pB125R protein fragments were analyzed by SDS-PAGE and Western blotting. (**C**,**D**) After the first round of identification, further truncation was performed for Western blotting analysis using mAb 9G11. (**E**) A single AA mutant protein identifies critical AAs. AA: Amino acid. (**F**) Western blotting analysis of pB125R epitope proteins recognized by mAb 9G11 was performed using anti-His antibody and ASFV-positive serum.

**Figure 5 viruses-16-01257-f005:**
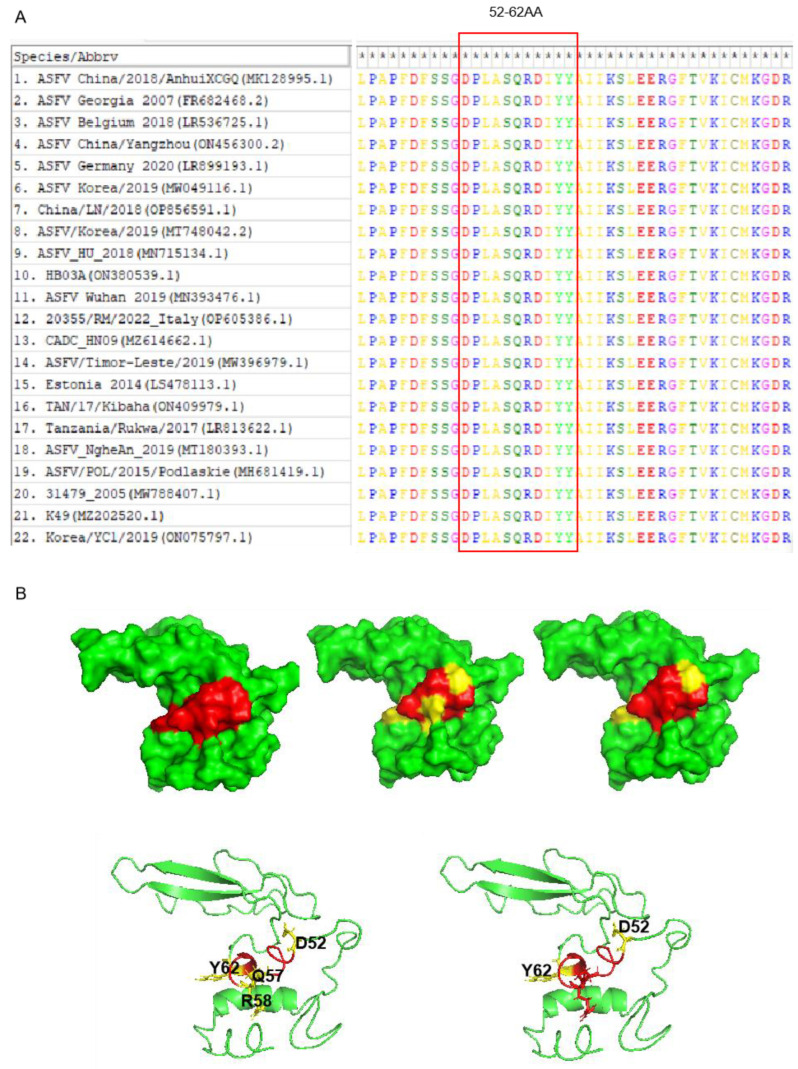
Bioinformatics analysis of pB125R protein epitope. (**A**) Analysis of conserved epitopes of pB125R gene. Comparison of epitope homology of ASFV pB125R (MK128995.1) with other ASFV strains in GenBank. (**B**) Spatial structure analysis of pB125R and spatial distribution of epitopes in pB125R recognized by mAb 9G11. Red: Linear amino acid sequence recognized by 9G11 monoclonal antibody; Yellow: Key amino acid site recognized by 9G11 monoclonal antibody.

**Figure 6 viruses-16-01257-f006:**
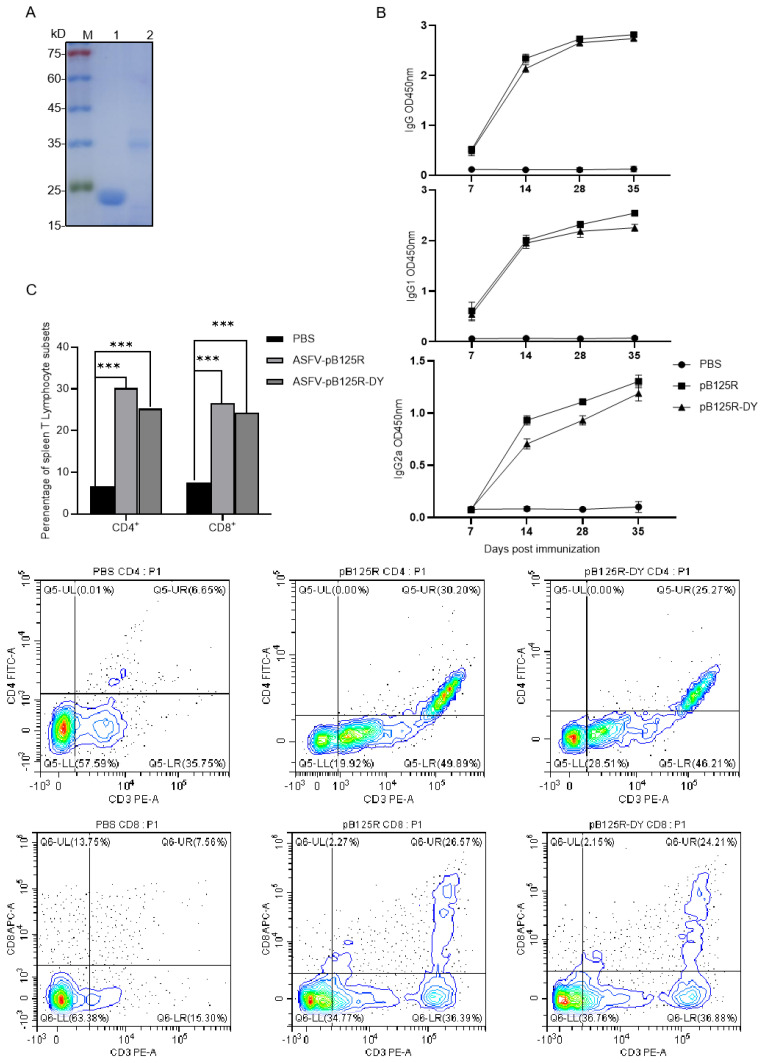
Epitope immunoactivity verification. (**A**) Purification of recombinant proteins. M: Protein molecular weight marker; lane 1: pB125R recombinant protein; lane 2: pB125R-DY epitope recombinant protein. (**B**) The dynamic changes in IgG, IgG1, and IgG2a antibodies in mice were detected. (**C**) T cell subsets were determined. Note: *p* < 0.001 ***.

**Table 1 viruses-16-01257-t001:** Sequences of primers used in pB125R prokaryotic expression plasmid.

Title	Primers	Sequences (5′–3′)
pB125R	F	ATCGGATCCGAATTCATGGCGGTTTATGCG
	R	GTGGTGGTGCTCGAGTCTAGACTCTAAAAATT

**Table 2 viruses-16-01257-t002:** Prokaryotic expression of the primer of truncated body pB125R.

Segment	Sequences (5′-3′)		Positions (Amino Acid)
pB125R-1	F	ATCGGATCCGAATTCATGGCGGTTTATGCG	1-40AA
	R	GTGGTGGTGCTCGAGATACACCAAAAAGTT	
pB125R-2	F	GTGGTGGTGCTCGAGATACACCAAAAAGTT	31-70AA
	R	GTGGTGGTGCTCGAGCTCCTCGAGGCTTTT	
pB125R-3	F	ATCGGATCCGAATTCTACTATGCCATCATA	61-100AA
	R	GTGGTGGTGCTCGAGTTTGTTTATCTCAAT	
PB125R-4	F	ATCGGATCCGAATTCAAAAAAATACAATCC	91-125AA
	R	GTGGTGGTGCTCGAGTCTAGACTCTAAAAA	
pB125R-2-1	F	AATTCAACTTTTTGGTGTATGAACTACCTGCCCCTTTTGACTTTTCCTCCC	36-50AA
	R	TCGAGGGAGGAAAAGTCAAAAGGGGCAGGTAGTTCATACACCAAAAAGTTG	
pB125R-2-2	F	AATTCGAACTACCTGCCCCTTTTGACTTTTCCTCCGGCGACCCTTTGGCCC	41-55AA
	R	TCGAGGGCCAAAGGGTCGCCGGAGGAAAAGTCAAAAGGGGCAGGTAGTTCG	
pB125R-2-3	F	AATTCTTTGACTTTTCCTCCGGCGACCCTTTGGCCAGTCAGCGCGACATAC	46-60AA
	R	TCGAGTATGTCGCGCTGACTGGCCAAAGGGTCGCCGGAGGAAAAGTCAAAG	
pB125R-2-4	F	AATTCGGCGACCCTTTGGCCAGTCAGCGCGACATATACTATGCCATCATAC	51-65AA
	R	TCGAGTATGATGGCATAGTATATGTCGCGCTGACTGGCCAAAGGGTCGCCG	
pB125R-2-5	F	AATTCGACCCTTTGGCCAGTCAGCGCGACATATACTATGCCATCC	52-64AA
	R	TCGAGGATGGCATAGTATATGTCGCGCTGACTGGCCAAAGGGTCG	
pB125R-2-5-1	F	AATTCCCTTTGGCCAGTCAGCGCGACATATACTATGCCATCC	53-64AA
	R	TCGAGGATGGCATAGTATATGTCGCGCTGACTGGCCAAAGGG	
pB125R-2-5-2	F	AATTCTTGGCCAGTCAGCGCGACATATACTATGCCATCC	54-64AA
	R	TCGAGGATGGCATAGTATATGTCGCGCTGACTGGCCAAG	
pB125R-2-5-3	F	AATTCGCCAGTCAGCGCGACATATACTATGCCATCC	55-64AA
	R	TCGAGGATGGCATAGTATATGTCGCGCTGACTGGCG	
pB125R-2-5-4	F	AATTCAGTCAGCGCGACATATACTATGCCATCC	56-64AA
	R	TCGAGGATGGCATAGTATATGTCGCGCTGACTG	
pB125R-2-5-5	F	AATTCCAGCGCGACATATACTATGCCATCC	57-64AA
	R	TCGAGGATGGCATAGTATATGTCGCGCTGG	
pB125R-2-5-6	F	AATTCCGCGACATATACTATGCCATCC	58-64AA
	R	TCGAGGATGGCATAGTATATGTCGCGG	
pB125R-2-5-7	F	AATTCGACATATACTATGCCATCC	59-64AA
	R	TCGAGGATGGCATAGTATATGTCG	
pB125R-2-5-8	F	AATTCATATACTATGCCATCC	60-64AA
	R	TCGAGGATGGCATAGTATATG	
pB125R-2-5-9	F	AATTCGACCCTTTGGCCAGTCAGCGCGACATATACTATGCCC	52-63AA
	R	TCGAGGGCATAGTATATGTCGCGCTGACTGGCCAAAGGGTCG	
pB125R-2-5-10	F	AATTCGACCCTTTGGCCAGTCAGCGCGACATATACTATC	52-62AA
	R	TCGAGATAGTATATGTCGCGCTGACTGGCCAAAGGGTCG	
pB125R-2-5-11	F	AATTCGACCCTTTGGCCAGTCAGCGCGACATATACC	52-61AA
	R	TCGAGGTATATGTCGCGCTGACTGGCCAAAGGGTCG	

**Table 3 viruses-16-01257-t003:** pB125R truncated protein amino acid sequence.

Title	Amino Acid Sequence	
pB125R-1	MAVYAKDLDNNKELNQKLINDQLKIIDTLLLAEKKNFLVY	1-40AA
pB125R-2	LAEKKNFLVYELPAPFDFSSGDPLASQRDIYYAIIKSLEE	31-70AA
pB125R-3	YYAIIKSLEERGFTVKICMKGDRALLFITWKKIQSIEINK	61-100AA
PB125R-4	KKIQSIEINKKEEYLRMHFIQDEEKAFYCKFLESR	91-125AA
pB125R-2-1	NFLVYELPAPFDFSS	36-50AA
pB125R-2-2	ELPAPFDFSSGDPLA	41-55AA
pB125R-2-3	FDFSSGDPLASQRDI	46-60AA
pB125R-2-4	GDPLASQRDIYYAII	51-65AA
pB125R-2-5	DPLASQRDIYYAI	52-64AA
pB125R-2-5-1	PLASQRDIYYAI	53-64AA
pB125R-2-5-2	LASQRDIYYAI	54-64AA
pB125R-2-5-3	ASQRDIYYAI	55-64AA
pB125R-2-5-4	SQRDIYYAI	56-64AA
pB125R-2-5-5	QRDIYYAI	57-64AA
pB125R-2-5-6	RDIYYAI	58-64AA
pB125R-2-5-7	DIYYAI	59-64AA
pB125R-2-5-8	IYYAI	60-64AA
pB125R-2-5-9	DPLASQRDIYYA	52-63AA
pB125R-2-5-10	DPLASQRDIYY	52-62AA
pB125R-2-5-11	DPLASQRDIY	52-61AA

**Table 4 viruses-16-01257-t004:** ASFV strains collected from NCBI and used to align the sequences of the identified epitopes.

Accession Number	Title	Country	Collection Date	Lengths of pB125R(AA)
MK128995.1	ASFV China/2018/AnhuiXCGQ	China	2021	125
FR682468.2	ASFV Georgia 2007	UK	2020	125
LR536725.1	ASFV Belgium 2018	Germany	2019	125
ON456300.2	ASFV China/Yangzhou	China	2023	125
LR899193.1	ASFV Germany 2020	Germany	2020	125
MW049116.1	ASFV Korea/2019	Korea	2022	125
OP856591.1	China/LN/2018	China	2023	125
MT748042.2	ASFV/Korea/2019	Korea	2023	125
MN715134.1	ASFV_HU_2018	Hungary	2021	125
ON380539.1	HB03A	China	2022	125
MN393476.1	ASFV Wuhan 2019	China	2020	125
OP605386.1	20355/RM/2022_Italy	Italy	2023	125
MZ614662.1	CADC_HN09	China	2021	125
MW396979.1	ASFV/Timor-Leste/2019	Australia	2022	125
LS478113.1	Estonia 2014	Germany	2018	125
ON409979.1	TAN/17/Kibaha	Tanzania	2023	125
LR813622.1	Tanzania/Rukwa/2017	Kenya	2020	125
MT180393.1	ASFV_NgheAn_2019	S. Korea	2021	125
MH681419.1	ASFV/POL/2015/Podlaskie	Denmark	2019	125
MW788407.1	31479_2005	Italy	2021	125
MZ202520.1	K49	Russia	2021	125
ON075797.1	Korea/YC1/2019	Korea	2022	125

## Data Availability

The data analyzed during the current study are available from the corresponding author upon reasonable request.

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
