# Peer review of "Identification and Characterization of a Novel B Cell Epitope of ASFV Virulence Protein B125R Monoclonal Antibody"

_viruses, 2024, doi:10.3390/v16081257_

Round 1

Reviewer 1 Report (New Reviewer)

Comments and Suggestions for Authors

Dear editor

The manuscript entitled "Identification and characterization of a novel B cell epitope of ASFV virulence protein B125R monoclonal antibody" by Dr. Zhao et al., represents another research that has the aim of identifying one protein encoded by ASFV genome, and analyze the ability of this protein to induce the immune system. 

My evaluation of the paper is that it can be accepted. 

Comments on the Quality of English Language

Another check of the English language can be beneficial for the paper quality 

Author Response

Comments 1:

The manuscript entitled "Identification and characterization of a novel B cell epitope of ASFV virulence protein B125R monoclonal antibody" by Dr. Zhao et al., represents another research that has the aim of identifying one protein encoded by ASFV genome, and analyze the ability of this protein to induce the immune system.

My evaluation of the paper is that it can be accepted.

Another check of the English language can be beneficial for the paper quality.

Response 1:

Thank you for your review and evaluation of this manuscript. In order to improve the English quality of this manuscript, we have re-checked the full text and corrected some inappropriate places in the manuscript, which has been supplemented and marked, please review it again.

Reviewer 2 Report (New Reviewer)

Comments and Suggestions for Authors

In this article, the authors describe the production of monoclonal antibodies against the B125R protein of the African swine fever virus.  Antigenic epitopes of B125R protein, recognized by these antibodies, have also been identified.

These results can be used for serological tests for African swine fever, but even for this it is necessary to conduct studies of serum obtained from infected animals. The authors mention only one experiment using swine serum and a full-size protein from E. coli. At the same time, there is no control in this experiment: western blotting with recombinant protein and serum from pigs negative for ASFV. The recognition of the identified epitope by the serum of infected animals has also not been confirmed.

The use of the obtained results for the development of subunit vaccines is extremely doubtful, since so far the presence of virus neutralizing antibodies against the ASFV has not been confirmed. To study the T-cell response, it is also necessary to use pig cells in experiments, not mouse cells.

Comments on the Quality of English Language

Unfortunately, the manuscript is difficult to read due to incorrect grammatical construction of sentences, as well as incorrect use of terminology when describing experiments and results. I would recommend that the authors show the manuscript to native speakers.

Author Response

Comments 1:

In this article, the authors describe the production of monoclonal antibodies against the B125R protein of the African swine fever virus.  Antigenic epitopes of B125R protein, recognized by these antibodies, have also been identified.

These results can be used for serological tests for African swine fever, but even for this it is necessary to conduct studies of serum obtained from infected animals. The authors mention only one experiment using swine serum and a full-size protein from E. coli. At the same time, there is no control in this experiment: western blotting with recombinant protein and serum from pigs negative for ASFV. The recognition of the identified epitope by the serum of infected animals has also not been confirmed.

The use of the obtained results for the development of subunit vaccines is extremely doubtful, since so far the presence of virus neutralizing antibodies against the ASFV has not been confirmed. To study the T-cell response, it is also necessary to use pig cells in experiments, not mouse cells.

Unfortunately, the manuscript is difficult to read due to incorrect grammatical construction of sentences, as well as incorrect use of terminology when describing experiments and results. I would recommend that the authors show the manuscript to native speakers.

Response 1:

We greatly appreciate your review and professional feedback on this manuscript, and as you are concerned, there are still some issues in the article that need to be resolved.

In Figure 1, we took the induced blank carrier protein as a negative control, and used His labeled antibody and ASFV positive serum to detect whether recombinant pB125R was successfully expressed and had structural characteristics similar to those of natural viruses. The results were in line with the expected results, indicating that the recombinant protein we prepared could be used for subsequent studies.

We added the data in Figure 4F to confirm that the pB125R epitope protein screened in this study had good reaction characteristics with ASFV-positive serum.

We also know that the natural host of ASFV is pig, and it is best to use pig cells to study the effect of ASFV pB125R epitope protein on cell response. In this study, we used mouse cells to preliminary reveal that an epitope recognized by mAb 9G11 can induce an effective humoral and cellular immune response in a mouse model. However, the experimental cost of pig cell samples is high, and the experimental design needs to be more rigorous. In follow-up studies, we will refer to the current preliminary conclusions to carry out studies on the effects on pig cells.

Thank you again for your review and suggestions on this manuscript. We have tried our best to explain the questions you raised and supplement the experimental data. At the same time, we re-examined the manuscript, revisions were implemented in revision mode, and corrected some inappropriate or irregular parts. I hope you can consider the revised manuscript.

Round 2

Reviewer 2 Report (New Reviewer)

Comments and Suggestions for Authors

I don't have any more comments.

This manuscript is a resubmission of an earlier submission. The following is a list of the peer review reports and author responses from that submission.

Round 1

Reviewer 1 Report

Comments and Suggestions for Authors

The data presented in the manuscript: "Identification and characterization of a novel B cell epitope of ASFV virulence protein B125R monoclonal antibody" showed the identification of B cell epitope pB215R-DY and that could induce effective humoral and cellular immune responses in mice.

Although all the in-vitro data showed, like the pB215R protein expression and purification, the production of the monoclonal antibodies, the identification of the heavy and light chains of IgGs, etc., the importance losses power, when the authors does not include an experiment with serum pig samples. Due the fact that this is a disease that affects only pigs, will be interesting to do this experiment. Same with the immunization of mice. The authors should include that experiment in here.

Author Response

Thank you for your valuable advice. We also know that the natural host of ASFV is pigs, and the detection of ASFV pB125R epitope protein is of great research significance for the study of pig serum samples. In this study, we conducted a preliminary investigation on mAb 9G11, which can better recognize pB125R, and the results of this investigation preliminarily revealed that the epitope recognized by mAb 9G11 can induce effective humoral and cellular immune responses in mouse models. However, the cost of pig serum sample experiment is high, and more rigorous experimental design is still needed. In the follow-up studies, we will refer to the current preliminary conclusions and carry out studies on the effects of ASFV pB125R on pigs.

Reviewer 2 Report

Comments and Suggestions for Authors

In this manuscript, the authors reported two monoclonal antibodies (mAbs) against the pB125R named 9G11 and 15A9. The antigenic epitope recognized by mAb 9G11 was the 52DPLASQRDIYY62 on ASFV pB125R and this epitope was highly conserved among different strains. The key amino acids were identified as D52, Y62, Q57, and R58 by single-point mutation of 11 amino acids of the epitope by alanine scanning. In addition, the immunological effects of the epitope (pB125R-DY) against 9G11 were evaluated in mice, and the results showed that both full-length pB125R and the epitope pB125R-DY could induce effective humoral and cellular immune responses in mice.

Comments:

1. M&M

Details of serum samples used in animal experiments should be included. Procedures for mouse inoculation with rec proteins and serum collection should be added.

2. Fig 1. (C) Western blotting analysis of recombinant pB125R protein using 206 ASFV-positive serum.

Please include the information of the positive serum used, such as collected from field or experimental animals, and collected on which days of infection.

3. Line 215-218

Do the mouse sera react with ASV virus particles?

4. Fig 2a

Please indicate what negative serum was used.

5. Fig 2b

Please provide positive serum information.

6. Fig 6b

How many mouse sera are the averages derived from?

7. Line 318: “These two monoclonal antibodies could both react with the eukaryotic expressed pB125R protein and ASFV positive serum,

How do monoclonal antibodies react with virus-positive sera?

8. Line 319: “indicating that the monoclonal antibodies can be used as good tools for the early detection of ASFV antigen infection.”

Do you have any data to support this comment? Do the mAbs react with ASF virus particles?

9. line 357-358  “this epitope contributes to the effective clearance of viral infections, and the epitope identified by mAb 9G11 acted as an immunogen to activate humoral and cellular immunity in immunized animals.

Please discuss how a leaner B cell epitope induced cellular immune response.

10. Line 364 “This epitope is highly conserved across strains.”

How many positive sera from different strains of ASFV were tested against the peptide and rec protein to confirm that this epitope is conserved? The serum information should be included in the manuscript.

11. Discussions are lengthy and aimless. Key points should be strengthened and highlighted.

Overall, this work is valuable. However, many comments and conclusions are not supported by experimental results. The manuscript can be brought to accuracy by carefully editing the content. In its current form, it is not suitable for publication in a peer-reviewed journal.

Comments on the Quality of English Language

The manuscript can be made more readable through careful editing of language. 

Reviewer 3 Report

Comments and Suggestions for Authors

ASF is a highly infectious disease that severely affects the swine industry. B125R encodes a non-structural protein that plays an important role during ASFV infection. In this study, the authors identified antigenic epitope of pB125R, and verified pB125R epitope performed the immunoactivity to promoted CD4+ and CD8+ T cell metabolism in mice. However, before considering publication, there are some mistakes that need to be corrected added.

1.     In Fig 4C, the epitope of pB125R-51-65AA contains the epitope of pB125R-52-64AA, but its immunoactivity is not as strong as that of pB125R-52-64AA against mAb 9G1.

2.     In Fig 6B, is there an extra "A"?

3.     In Fig 6C, what cell markers are labeled by APC, PE and FITC respectively? Can they be shown on the figure?

Comments on the Quality of English Language

There are some written errors, such as Line 21 (an extra parenthesis), Line 184 (106/mL).
